# Experiences and perceptions of trial participants and healthcare professionals in the UK Frozen Shoulder Trial (UK FROST): a nested qualitative study

Cynthia Srikesavan [1], Francine Toye,[2] Stephen Brealey,[3] Lorna Goodchild,[4] Matthew Northgraves,[5] Charalambos P Charalambous,[6,7] Amar Rangan [3], Sarah Lamb[1,8]

For numbered affiliations see end of article.

**Correspondence to**
Dr Francine Toye;
Francine.Toye@ouh.nhs.uk

## ABSTRACT

**Objectives** To explore the experiences and perceptions of trial participants and healthcare professionals in the UK Frozen Shoulder Trial (UK FROST), a multicentre randomised controlled trial that compared manipulation under anaesthesia (MUA), arthroscopic capsular release (ACR) with a 12-week early structured physiotherapy programme (ESP) in people with unilateral frozen shoulder referred to secondary care.

**Design** Nested qualitative study with semistructured interviews. We used constant comparison method to develop our themes.

**Setting** This qualitative study was nested within the UK FROST.

**Participants** 44 trial participants (ESP: 14; MUA: 15; ACR: 15), and 8 surgeons and 8 physiotherapists who delivered the treatments in the trial.

**Results** Trial participants found UK FROST treatments acceptable and satisfactory in terms of content, delivery and treatment benefits. Participants in all arms experienced improvements in pain, shoulder movements, and function. Participants said they would choose the same treatment that they received in the trial. Surgeons and physiotherapists felt that the content and delivery of UK FROST treatments was not significantly different to their routine practice except for the additional number of physiotherapy sessions offered in the trial. They had mixed feelings about the effectiveness of UK FROST treatments. Both stressed the value of including hydrodilatation as a comparator of other treatment options. Physiotherapists raised concerns about the capacity to deliver the number of UK FROST physiotherapy sessions in routine clinical settings.

Shared perceptions of trial participants, surgeons and physiotherapists were: (1) Pain relief and return of shoulder movements and function are important outcomes and (2) Adherence to exercises leads to better outcomes.

**Conclusion** In general, our findings indicated that trial participants, and surgeons and physiotherapists who delivered the treatments had positive experiences and perceptions in the UK FROST. Early qualitative investigations to explore the feasibility of delivering

## Strengths and limitations of this study

► Interviews were conducted and analysed by a researcher who was not involved in the UK Frozen Shoulder Trial (UK FROST).
► Interview codes and themes were reviewed by an experienced researcher to ensure rigour of analysis and interpretation of data.
► Interviews were conducted with trial participants and healthcare professionals who participated in the UK FROST. Therefore, the findings may not be transferable outside this context.
► Ninety-three per cent of the interviews were telephone based. It is uncertain if interviewees would have expressed differently in face-to-face interviews.
► Those who did not participate in the interviews may have had different experiences.

treatments in real-world settings are suggested in future trials in the frozen shoulder.

**Trial registration number** International Standard Randomised Controlled Trial Register, ID: ISRCTN48804508. Registered on 25 July 2014; Results

## INTRODUCTION

Frozen shoulder or adhesive capsulitis is a chronic shoulder condition commonly affecting people aged between 50 and 60 years.[1] Women are affected four times than men,[2] while the prevalence rate is around 20% in people with diabetes.[2] Frozen shoulder is characterised by three clinical stages that often overlap[3 4]: (1) a painful stage (3–8 months), (2) a stiffness (frozen) stage (4–12 months) with progressive loss of active and passive shoulder movements and (3) a thawing stage (12–42 months) with gradual recovery of movements in most people.[2]

Frozen shoulder is managed by non-surgical or surgical treatments.[3 5–9] However, there is no high-quality evidence to recommend one treatment over another.[3] The UK Frozen Shoulder Trial (UK FROST) is a multicentre, pragmatic randomised controlled trial in adults with frozen shoulder.[10] It evaluated the clinical and cost-effectiveness of two common surgical treatments: manipulation under anaesthesia (MUA) and arthroscopic capsular release (ACR) both followed by up to 12 weeks of postprocedural physiotherapy (PPP), with a 12-week early structured physiotherapy (ESP) programme that included an intra-articular steroid injection. Detailed descriptions of the trial treatments, including the specifically developed ESP and PPP pathways are available.[10]

The qualitative literature on frozen shoulder is limited.[11 12] There are no trials in frozen shoulder with an embedded qualitative component that explored the perspectives of trial participants (patients with frozen shoulder) and healthcare professionals who delivered the trial treatments. Therefore, we proposed a nested qualitative study within the UK FROST to explore the experiences and perceptions of trial participants who received UK FROST treatments, and healthcare professionals (surgeons and physiotherapists) who delivered those treatments.

## METHODS

A qualitative and constructivist approach[13] was adopted to understand the experiences and perceptions of trial participants and healthcare professionals in the UK FROST.

The primary author (CS) is a physiotherapy researcher trained in qualitative methods and not involved in the delivery of UK FROST treatments and outcome measurement. CS conducted all the interviews and led the analysis. FT is an experienced qualitative researcher, anthropologist and physiotherapist, also not involved in delivery of care or outcome measurement. FT played a collaborative role in analysis.

Participants who took part in the main trial had primary unilateral frozen shoulder, identified through a restriction of passive external rotation on clinical examination. They were randomly assigned to MUA, ACR or ESP treatment arms in the ratio of 2:2:1. A purposive sample of trial participants was recruited from those who had agreed to be contacted by the trial team for interviews at approximately 12 months postrandomisation, which coincided with the primary endpoint of the trial and completion of UK FROST treatments.[10] We proposed to recruit up to 45 trial participants. As gender and diabetes are likely to have an impact on the treatment outcomes,[2 10] we also planned to include an equal number of men and women, and those with and without diabetes.

We also planned to recruit a purposive sample of up to 15 surgeons and physiotherapists who delivered the treatments in the trial and agreed to take part in the interviews.

The study information sheet and consent form were posted to potential trial participants and emailed to surgeons and physiotherapists. Two postal or email reminders were sent to those who did not respond within 4weeks of invitation. Once signed consent was received via post or email, CS contacted potential participants via telephone to coordinate an interview appointment with them.

Separate semistructured interview guides with open-ended questions were used for trial participants and health professionals. Given the geographical spread of those participated, interviews were conducted face to face or via telephone. Interviews were audiorecorded using a digital recorder, anonymised and transcribed verbatim by a professional transcription agency.

The interviews were analysed using constant comparative method[13 14] to develop categories and themes with a shared essence. The QSR International's NVivo V.11 software[15] was used to organise and assist analysis of the data. The analysis was stopped at the point where additional data no longer contributed to the understanding of the themes. To enhance credibility of the findings, codes and themes were reviewed by an experienced qualitative researcher (FT). The Standards for Reporting Qualitative Research (SRQR) guidelines were used to report the study.[16]

### Patient and public involvement
The patient interview guide was developed following discussions with patients with frozen shoulder, research team, a physiotherapist and surgeon with expertise in this area, and reviewing the literature.

### RESULTS
Sixty interviews (participants: 44; surgeons: 8 and physiotherapists: 8) were completed between August 2016 and January 2018. Most interviews (56/60, 93%) were conducted via telephone. Four interviews were held at the clinic offices of surgeons and physiotherapists. Though constant comparison was used to analyse data, a formal theory was not developed.

#### Interviews with trial participants
A flow diagram showing the number of patients invited, excluded and participated in the interviews is presented in figure 1. Participant characteristics are presented in table 1. Apart from two participants, all other participants received their allocated treatment.

Five themes were identified. The themes were not different between men and women; and between those with and without diabetes.

#### Experiences and perceptions of participating in the UK FROST
In general, trial participants had positive experiences in participating in UK FROST. They described altruistic and personal reasons for taking part in the trial. They were concerned about their upper limb disability from

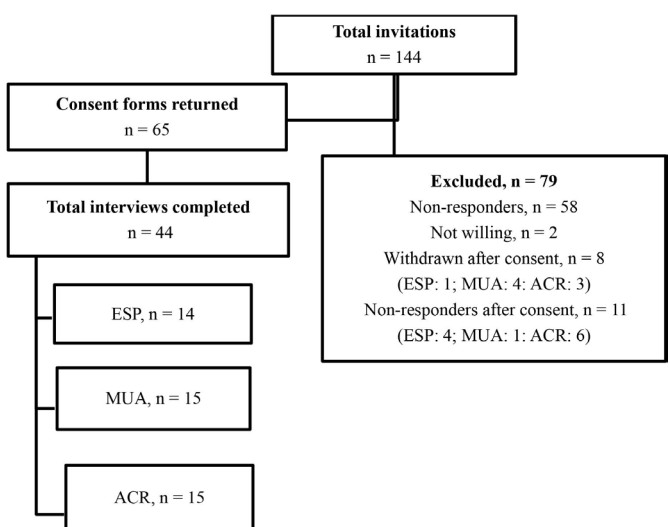

**Figure 1** Interviews with trial participants. ACR, arthroscopic capsular release; ESP, early structured physiotherapy; MUA, manipulation under anaesthesia.

frozen shoulder and were keen on getting it sorted in UK FROST. There was also an overall sense that the trial questionnaires were relevant, simple and straightforward to complete.

> Because I'm all in favour, if you can do something to help other people not go through the misery that you've been through, and gone through, then I would do it. (ESP)

Trial participants felt that their treatment packages were well coordinated and did not require any modification. The surgical procedures were clearly explained and physiotherapists who delivered ESP and PPP were supportive and helpful. Participants also liked to see the same physiotherapist for developing good connections and rapport throughout the study period. However, some ESP and ACR arm participants felt that the exercises were time consuming and found it difficult to fit into their daily life. Some said that they would have liked more information about continuing exercises after the trial and exercising when in pain.

> Operation, I believe obviously went fantastic, went really well. It was just a case of waking up, recovery. (ACR)

**Table 1** Characteristics of the participants interviewed

| UK FROST arms | Male | Female | Age Median (IQR) in years | Diabetic | Non-diabetic |
|---|---|---|---|---|---|
| ESP | 5 | 9 | 58 (51–63.5) | 3 | 11 |
| MUA | 8 | 7 | 55 (53–57.5) | 5 | 10 |
| ACR | 7 | 8 | 59 (53–69) | 5 | 10 |

ACR, arthroscopic capsular release; ESP, early structured physiotherapy; IQR, Interquartile Range; MUA, manipulation under anaesthesia; UK FROST, UK Frozen Shoulder Trial.

Well personally I found it quite difficult to make sure I had them that many times in the day. I'm not too sure how somebody working or having a family could actually manage to fit it in because as I say, by the time going towards the middle to the end of the programme… (ESP)

### Acceptability: treatment satisfaction and improved outcomes

All UK FROST treatments were found to be acceptable, satisfactory and beneficial. Except for two participants in the ESP arm, others were satisfied with the UK FROST treatments they received. Of these two exceptions, one did not improve after physiotherapy in the trial and the other was not satisfied when the exercise sessions were supervised by an unfamiliar physiotherapist.

> I'm absolutely delighted with the treatment that I was given. I feel as though it did everything that I wanted it to do and expected it to do. (ESP)
> Very satisfied. I have no complaints at all (ACR).

Participants considered pain relief and return of shoulder movements and function as important treatment outcomes. Participants in all arms experienced improvements in pain, movements and function. In spite of achieving pain relief and improved function, some participants said they experienced mild and occasional pain and restrictions during certain end-range activities. However, this did not impact their daily functioning.

#### Pain

Trial participants experienced a substantial reduction in pain in all treatment arms. The ESP arm participants said that the steroid injections reduced pain and allowed them to start physiotherapy.

> When I went to the surgeon I was injected into my shoulder and the pain down my arm that more or less went straightaway. (ESP)
> So at the beginning I said the pain was ten and now after all my physios, I'd say it was, I'd say it was about two now. (MUA)
> I mean the pain in the beginning was just horrendous, it was really, really sore, really painful but after I'd had the physiotherapy, it was… I've got no pain at all now. (ACR)

#### Movements

Trial participants described how the physiotherapy sessions (ESP and PPP) had helped to improve their shoulder movements.

> I could tell initially straightaway that my movement was starting to come, within a few days I could tell a difference of doing the exercises and as the weeks went on, it was just got better and better and by the time the twelve weeks was up, I virtually had full movements with no pain or anything, it was brilliant! (ESP)

After a few days I was doing my exercises and I was quite surprised already how much movement I had back and then it was regular physio appointments up at the hospital just to keep moving things around and that went really well…the physiotherapy was actually really, really beneficial. (MUA)

In the ACR arm, improvement in movements was thought to be quicker than participants had expected. Some experienced improvements as early as 1–2 weeks of physiotherapy after surgery.

I felt that the physiotherapy I received was marvellous and improved the range of movements or showed me how to keep that range of movements much quicker than they did on the right-hand side, so I felt that everything went along fine, and I've got no complaints at all, none. (ACR)

It is almost like you have had a quick fix to fix your shoulder then you move on and I think personally for me because the surgery went very well and almost after a couple of weeks I was back to normal. (ACR)

### Function

Trial participants in all treatment arms described how their ability to do routine activities improved.

I can lift my arm above my head now, you know? I can carry stuff, and I can lift it above my waist, and I can actually go swimming, you know? I can swim now. (ESP)

My little everyday things have come back; I have come back, yes. (MUA)

I can do everything—there's nothing that I can't do; I can wash my back, I can put my bra on, fasten it at the back, I can fasten my skirt at the side and the back now, there's nothing I can't do before I had the frozen shoulder everything I could do then I can now do again. (ACR)

### Adherence to exercises

Trial participants described that ESP and PPP were difficult to begin with, but became easier on subsequent sessions.

It (Physiotherapy) was agony but from every session there was slight improvement, so for every kind, every session I had, I had more movement and less pain, so it got to a point where some days I would go down and I had no pain at all. (ESP)

However, participants were aware of the benefits of exercises and persevered with their home exercises during the study period.

It was uncomfortable but I tried really hard with them to be honest because it was so awful when I couldn't use my arm that I knew that the only way to help it was to do them as much as I could. (MUA)

Several participants said they did not continue their home exercises after the trial because they felt they had regained adequate shoulder function. Instead, they maintained their shoulder mobility by being functionally active. A few participants occasionally did some shoulder stretches.

I'm working with my shoulder all the time so I'm not doing the exercises that the hospital gave me, because I'm working my, I'm swimming, I'm doing…I go on long walks, I take the dog out and what have you, so I'm using my arm. (ESP)

### Personal treatment preferences

Trial participants had mixed treatment preferences before the trial. Some considered physiotherapy to be ineffective, while a few wanted to avoid the risks of surgery. Some preferred the less invasive MUA while some thought ACR as the final solution. Some didn't have any preference at all.

I'm not really too sure why I wouldn't choose physiotherapy. I just think surgery seems a more final option. Physiotherapy, it might work, it may help, it may not. But to me, if I was given surgery, the surgery would work. I had more faith in the surgery working than the physiotherapy itself. (ACR)

Well I didn't want to go to surgery or anything like that, so I just had the needle in my shoulder. I don't think I'd have wanted to go surgery at my age (ESP)

Despite preferences at the outset, at the end of the trial, there was a sense that participants would choose the same treatment they received in the UK FROST. In retrospect, some participants in ESP and MUA said that they would choose surgery as a permanent and time-saving solution for their shoulder problem.

Oh no I would have the same treatment. As I say I was only there for ten weeks and I mean, on my eleventh week I was still hell of a lot better. (ESP)

### Interviews with surgeons and physiotherapists

A flow diagram showing the number of health professionals invited, excluded and participated in the interviews is presented in figure 2. The characteristics of surgeons and physiotherapists are presented in table 2.

Four themes were identified.

### Experiences and perceptions of delivering UK FROST treatments

In general, surgeons and physiotherapists had positive experiences in delivering the trial treatments. Surgeons described how the surgical treatments were no different from their usual practice. They described how trial participants who received MUA or ACR were usually seen by specialised physiotherapists for advice and PPP at the follow-up stage. Some said there were no reports of complications or side effects due to surgery, while some said that a few patients did not respond well to ESP.

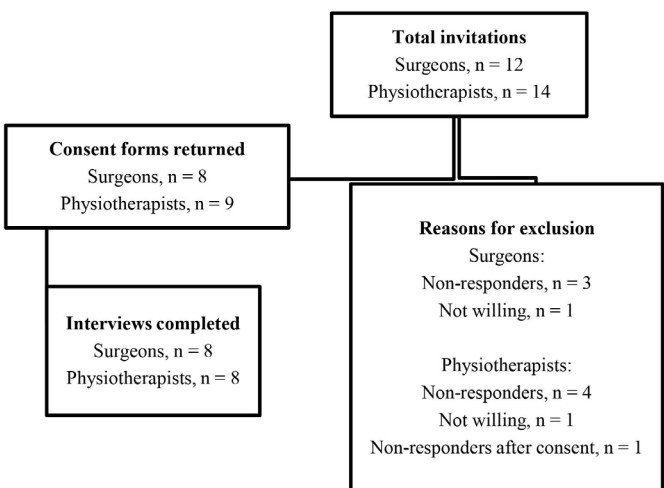

**Figure 2** Interviews with surgeons and physiotherapists.

Surgeon 1: Technically there was no difference, I wouldn't do anything different.

Surgeon 6: In the physiotherapy group, there have been a couple of patients who have not responded very well to surgery, very well to the treatment, to the physiotherapy treatment and we have just carried on and they've taken a bit longer.

Physiotherapists felt that the UK FROST physiotherapy programmes and the exercise booklet gave more flexibility in choosing the exercises than they would usually have in their routine practice. The only difference was the increased number of sessions offered in the trial. Surgical soreness was a commonly seen postoperative issue in ACR and MUA participants. However, it was quick to resolve with PPP compared with those who received ESP. There were a few suggestions from physiotherapists to improve the UK FROST exercise intervention. For example, one physiotherapist felt the exercises could have been standardised better. Two others suggested either spreading the 12 weekly sessions over 6 months or doing group sessions.

Physiotherapist 1: I think the interventions that were on the booklet were what I would use generally. There was always an option there for me to tick off what I would do so I was in agreement with the options that were there and in agreement with the options that they actually, didn't want you to use.

| Table 2 Characteristics of the surgeons and physiotherapists interviewed | | | |
|---|---|---|---|
| | **Gender** | | **Years of experience in treating shoulder conditions** |
| | **Male** | **Female** | **Median (IQR) years** |
| Surgeons | 7 | 1 | 11 (7–16.25) |
| Physiotherapists | - | 8 | 13.5 (12–15.75) |

IQR, Interquartile Range.

Physiotherapist 5: I think group sessions would be really useful because patients get a lot from each other, and having experience group sessions with other clients with different pathologies, you know, they find that really reassuring…

Physiotherapists also commented on the feasibility of delivering the UK FROST physiotherapy programmes within the National Health Service (NHS) settings. There was a sense that it would be difficult to deliver the number of UK FROST physiotherapy sessions in routine practice.

Physiotherapist 4: … they (Trial participants) were seen with the 24 hours' post-surgery and they had twelve sessions which is a luxury because in our Trust, that is never, not going to happen and that never used to happen.

### Expectations and preferences about UK FROST treatments
Although health professionals said that they maintained equipoise, some described mixed expectations regarding UK FROST treatments. For example, some felt that ESP would not be as effective as surgery.

Surgeon 7: My expectation is that physio won't work. My expectation is that the other two are probably Equivocal.

Some felt that ACR would perform better than ESP, while others felt that MUA would be comparable to ACR. There were also those who expected similar outcomes across arms.

Physiotherapist 6: I'm expecting the MUAs to be surprisingly better than I would expect. I think the arthrolysis do great anyway and the physio is an unfair one.

Surgeons and physiotherapists suggested hydrodilatation would be an easy to use, less invasive and inexpensive treatment and an appropriate alternative to combat NHS surgery waiting lists. They felt hydrodilatation should have been included as a potential comparator in UK FROST.

Surgeon 5: I'd definitely have a hydro-dilatation group because part of your trial is trying to work out if the cheaper operation is better than the more expensive operation and hydro-dilatations probably gained quite popularity since we started the trial design and it reflects current practice.

Physiotherapist 6: It needs hydro-dilatation in it. I personally think it gets really good results on a big bulk of trial participants and it's a wasted opportunity to have done this study and not have that as one of the arms.

### Factors that influence treatment outcomes
Similar to trial participants, surgeons and physiotherapists also felt that reduced pain, improved movement and function were important treatment outcomes. They considered that patient engagement and positive treatment

expectations would contribute to better outcomes, while diabetes would have a negative impact on prognosis.

> Physiotherapist 5: I think the expectations and belief is probably the most noticeable factor that affects people's outcome; if they believe something is the right thing, the best thing for them, they seem to do well.

> Surgeon 8: …I've seen plenty of them of male diabetics that say, 'Well, I've been stiff for 3 years,' or two and a half years, that's not uncommon

### Perceptions of trial participants' experience in the trial

Surgeons and physiotherapists described that some patients declined to take part in the UK FROST because they had previous physiotherapy that did not work and hence wanted to avoid randomisation into the ESP.

> Surgeon 8: Many of them they say, 'Look, I would love to contribute to the greater good and be involved in clinical trials, but I've come to the point that I will not consent for physiotherapy if I was randomised to that…

They also described that participants came with fixed ideas of what treatment they wanted in the trial.

> Physiotherapist 8: So, we saw a lot of frozen shoulders coming in, but a lot of them had fixed ideas of what treatment they wanted. They didn't want surgery yet, or they didn't want to take time off work was the other one, but less so. The standout one was they didn't want an operation, or they wanted to try physiotherapy and injection and then they would opt for surgery. They wanted it to be continuum like that, not a one or the other.

However, surgeons and physiotherapists felt that trial participants found all the treatment arms acceptable once they were in the trial.

> Surgeon 6: Once they had consented to be part of the study, and they had no problems because they were equally welcome on what treatment they would get… all those who once we enrolled them on to the study, were okay with that.

Physiotherapists described how trial participants were surprised with the number of PPP sessions they received.

> Physiotherapist 1: I think they've been grateful and surprised at the amount of treatment (PPP) that they are allowed to have following the procedure.

## DISCUSSION
### Principal findings

We interviewed the trial participants and healthcare professionals (surgeons and physiotherapists) to understand their experiences and perceptions of participating in the UK FROST. Our key findings were: Trial participants

reported improved treatment outcomes and satisfaction with the trial treatments. All those interviewed had their personal treatment preferences and expectations about UK FROST treatments. Exercise adherence was considered important for better treatment outcomes.

### Pain relief and return of shoulder movements and function are important treatment outcomes

Trial participants and healthcare professionals agreed that pain relief and improvement in function and movement were important treatment outcomes. Their priorities are similar to the results of a previous survey in 225 healthcare professionals in the UK[17] and three systematic reviews on shoulder outcomes.[18–20]

### Adherence to prescribed exercises is important for better outcomes

Trial participants and healthcare professionals described continued patient engagement with the prescribed exercise as important for better outcomes.[21–23] During the trial, UK FROST participants were motivated by treatment benefits[24] and did their home exercises regularly. They were also self-determined[24] to persevere through the pain associated with exercise. However, after the trial, participants did not continue exercising because they thought they achieved enough movement to allow adequate daily function. Therefore, they prioritised continuing their daily functional activities and ignored the occasional pain and mild restrictions during certain end-range movements. These findings are in line with a previous study which conceptualised participants' views on 'ideal' (no symptoms at all) and 'adequate' (return to function with residual deficits) recovery from musculoskeletal complaints.[25]

### Trial participants and healthcare professionals had their personal preferences/expectations about UK FROST treatments

It is natural and inevitable that patients have their own treatment preferences. It is well known that personal preferences have a psychological therapeutic impact on treatment outcomes.[26 27] If participants get their preferred treatment in a trial, they are more likely to stay motivated, comply better and respond well. Participants who do not receive their preferred treatment might lack motivation and become less engaged. Though our findings indicate that trial participants had mixed preferences before participation, all UK FROST treatments were well-received at the end.

Our interview findings also indicated mixed expectations and preferences among UK FROST surgeons and physiotherapists, mainly based on their clinical experience.[28–30] Expert surgical and physiotherapy experience require years of skill building, training and repeated practice. It would be highly unlikely if experts did not acquire personal expectations or preferences. Some might argue that this is integral to developing expertise. This has significant implications in trials as it has an important

impact on equipoise. Our findings confirm that personal treatment expectations do exist among healthcare professionals, especially when treatment decisions are mainly expertise-based rather than evidence-based in treatment decision making.

Surgeons and physiotherapists also perceived hydrodilatation as easy to administer, less invasive and a cost-effective treatment for frozen shoulder instead of surgery. These findings resonate with its growing popularity and usage in clinical practice,[31] in spite of a lack of sufficient evidence on efficacy and safety.[32][33]

## Strengths and limitations

Interviews were conducted by a researcher not involved in UK FROST and by using open-ended questions that allowed interviewees to express their opinions freely. The interview codes and themes were reviewed by an experienced qualitative researcher to ensure rigour of analysis and interpretation of data. We also discussed our preliminary findings with a team of physiotherapists who treat people with frozen shoulder and the UK FROST team to produce our final themes.'

This study has some limitations. Interviews were conducted with trial participants and healthcare professionals who participated in the UK FROST. Therefore, the findings may not be transferable outside this context. Second, only two trial participants who did not receive their allocated treatments took part. This could have influenced their predominantly positive experiences with the trial treatments. Third, 93% of the interviews were conducted via telephone. Therefore, we are uncertain if interviewees would have expressed differently in face-to-face interviews. Lastly, those who did not participate in the interviews may have had different experiences.

## Implications for clinical practice

All UK FROST treatments were perceived as acceptable, beneficial, and satisfactory. The benefits and anticipated risks of these treatments must be clearly communicated to patients during shared treatment decision making.

## Implications for future research

1. Trial participants and healthcare professionals have preferences for treatments. Future studies should aim to understand how these preferences may influence trial results.
2. Implementation of UK FROST physiotherapy programmes within different NHS settings may be challenging. Pilot implementation studies to identify scaling-up strategies would be useful.
3. More large-scale and high-quality randomised controlled trials to ensure clinical effectiveness and safety of hydrodilatation are required to guide evidence-based practice.

## CONCLUSION

This qualitative study has provided a fuller understanding of the perspectives of UK FROST trial participants and healthcare professionals. Future trial designs would usefully consider including qualitative research as part of intervention development to ensure feasibility of delivering treatments within the NHS and to guide implementation beyond the trial. Evaluation of hydrodilatation compared with other treatment options is favoured by healthcare professionals. A prognostic tool to assist timely advice and treatment for people at risk of poorer outcomes would be helpful.

**Author affiliations**
[1]Nuffield Department of Orthopaedics, Rheumatology, and Musculoskeletal Sciences, University of Oxford, Oxford, UK
[2]Physiotherapy Research Unit, Oxford University Hospitals NHS Foundation Trust Nuffield Orthopaedic Centre, Oxford, UK
[3]York Trials Unit, ARRC Building, Department of Health Sciences, University of York, York, UK
[4]The James Cook University Hospital, South Tees Hospitals NHS Foundation Trust, Middlesbrough, UK
[5]Hull York Medical School, Hull, UK
[6]Blackpool Teaching Hospitals NHS Foundation Trust, Blackpool, UK
[7]School of Medicine, University of Central Lancashire, Preston, UK
[8]College of Medicine and Health, University of Exeter, Exeter, UK

**Acknowledgements** The authors thank the patient contributors who took part in developing the interview guide and patients and healthcare professionals participated in the interviews. This work was supported by the National Institute for Health Research (NIHR) Oxford Biomedical Research Centre (BRC). The UK FROST team acknowledges the support of the NIHR Clinical Research Network.

**Contributors** CS and FT led on the study and drafted the manuscript. MN and SB contributed to recruitment. LG and SL provided expertise as physiotherapists. AR and CPC provided expertise as orthopaedic surgeons. All authors read, provided feedback and approved the final version.

**Funding** This qualitative work was funded as part of the UK FROST Trial by the National Institute for Health Research (NIHR) Health Technology Assessment programme (Project reference 13/26/01).

**Competing interests** CS, FT, CPC declare no competing interests. LG, SB, MN report receiving grant from NIHR Health Technology Assessment programme that funded the trial. SL reports grants from the NIHR Health Technology Assessment Programme during the conduct of the study and was a member of several HTA boards from 2010 to 2015 and the NIHR CTU Standing Advisory Committee 2012–2016. AR reports other grants from NIHR HTA, ORUK and H2020 during the conduct of this study; and is a member of the NIHR i4i Challenge Awards Committee (2019-current). South Tees Hospitals NHS Foundation Trust receives educational grant to the department from DePuy Synthes (J&J). Institution also receives payment from DePuy Synthes (J&J) for Professor Rangan as the coordinating Investigator for the GLOBAL ICON Stemless Shoulder System Post Market Clinical Follow up Study: CT 1401. These are outside and unrelated to the submitted work.

**Patient and public involvement** Patients and/or the public were involved in the design, or conduct, or reporting, or dissemination plans of this research. Refer to the Methods section for further details.

**Patient consent for publication** Not required.

**Ethics approval** Ethics approval was obtained from the NRES Committee North East -Newcastle and North Tyneside 2 ethics committee (14/NE/1176) and Health Research Authority (HRA).

**Provenance and peer review** Not commissioned; externally peer reviewed.

**Data availability statement** All data relevant to the study findings are included in the article.

**ORCID iDs**
Cynthia Srikesavan http://orcid.org/0000-0002-3540-8052
Amar Rangan http://orcid.org/0000-0002-5452-8578

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
