## [Reviewer comments · BMJ Open]

ARTICLE DETAILS

TITLE (PROVISIONAL)	The experiences and perceptions of trial participants and healthcare professionals in the United Kingdom Frozen Shoulder Trial (UK FROST): A nested qualitative study.
AUTHORS	Srikesavan, Cynthia; Toye, Francine; Brealey, Stephen; Goodchild, Lorna; Northgraves, Matthew; Charalambous, C; Rangan, Amar; Lamb, Sarah

VERSION 1 – REVIEW

REVIEWER	Giovanni Galeoto Sapienza University of Rome
REVIEW RETURNED	30-Aug-2020

GENERAL COMMENTS	The article is well constructed and consistent with international guidelines. I ask the authors to cite a work that could be useful for the introduction and discussion of the manuscript: Cavalieri, E., Servadio, A., Berardi, A., Tofani, M., & Galeoto, G. (2020). The Effectiveness of Physiotherapy in Idiopathic or Primary Frozen Shoulder: a Systematic Review and Meta-Analysis. Muscles, Ligaments & Tendons Journal (MLTJ) , 10 (1).
---

REVIEWER	Glen Rae Sunderland and South Tyneside Foundation Trust, UK
REVIEW RETURNED	03-Sep-2020

GENERAL COMMENTS	The reviewer completed the checklist but made no further comments.
--

REVIEWER	Dr Claire Diver School of Health Sciences University of Nottingham England UK
REVIEW RETURNED	03-Dec-2020

GENERAL COMMENTS	Whilst I have some points to raise that I think need addressing I found this study interesting, relevant and well written. I would support a re-submission. Abstract The methods in the abstract do not detail the method of data analysis. The results in the abstract and main body of text report on participant perceptions; if perceptions are a phenomena to be explored then this should be reflected in the title and study aims as perceptions are a different phenomena to experiences.
--

	The conclusion in the abstract does not state state what 'important insights' have been gained and as it stands lacks specificity to this study. Introduction Does not adequately provide the gap in the literature or rationale for this study of FROST trial experience. p6 line 19 states little is known about the patient experience of frozen shoulder; this study however does not fill this gap as it is focussed on trial experience rather than condition experience. p6 line 29 states this was a study to look at experience and ACCEPTABILITY. the authors would be encouraged to be consistent throughout including the title, abstract and full text as to the aim of this study. Method pg6 line 40. Patients with a diagnosis of frozen shoulder were included in the study; can you state what the diagnostic criteria was for frozen shoulder as this is not given pg 4 line 42. It is not clear why participants were recruited at the end of the trial rather than during it; can you justify. p6 line 46. There is no justification for the sample size e.g. pragmatic, data saturation or information power. Can this be added. nor the method of sampling used e.g. purposive, convenience etc pg 6 line 50; the professional identity of the researcher conducting the interviews is not stated; this could affect the data collected and its analysis pg6 line 52 there is no justification for the use of a qualitative methodological approach or the use of semi-structured interviews. Can this be added. it is not clear whether there was a separate interview guide for patient participants and the health care professionals. This needs stating. p6 line 54 It is not stated who transcribed the interviews; for transparency can this be stated. There is no mention of an information sheet being provided to participants or how the approach to health care professionals was made. Results Throughout the results the description of the theme is limited. I would expect to see this expanded and more discrete use of quotes to demonstrate the validity of the description. The multiple quotes for one sub-theme often add little more of value to their presence in the data. I would suggest a re-write of the results with this in mind. The theme 'participation in UK FROST has the level of description I would like to see for each theme; this is in contrast to theme 2 Acceptability of treatment which is dominated by quotes rather than description of the theme. p8 Theme 1 Living with frozen shoulder. The aim of the study is to explore the experiences of trial participation. This theme does not do this, but touches on reasons for trial participation. I would suggest this theme is therefore removed as you describe trial participation as a separate theme. p11 line 46 'a stage based approach to routine clinical practice'. As with theme 1 for patient participants this theme does not describe the experiences of delivering treatment in the context of the FROST trial. I would suggest that this is therefore removed and the emphasis is on the delivery of treatment in the context of how it was delivered in FROST, assuming you have the data. This would have a slightly emphasis than theme 2 for this group ie theme 1 is about delivery of treatment in FROST and theme 2 about their expectations and preferences.
--	---

	Discussion As with the results there is a discussion of living with frozen shoulder that is outside the aim of the study which is to understand the experiences of participation in the trial itself. I would therefore suggest this is re-written and there is less emphasis on living with frozen shoulder and treatment in routine practice. Some conclusions are drawn that I do not see embedded in the results e.g. steroids help i can only seen mentioned once in patients theme 2 where receiving an injection is mentioned, and is in health care professionals approach in routine clinical practice rather than in the context of this trial Strengths and Limitatoin You acknowledge that findings cannot be extrapolated outside of the context of the FROST trial and yet some of your findings and areas for discussion are indicative of phenomena that occur outside of the trial eg routine practice, and living with frozen shoulder. If this study is about the FROST trial I would suggest that the emphasis remains there. This also links with: implications for practice. I feel that with the aim of the study to look at experiences of participating in the FROST trial the implications should be less about clinical practice and more about future research design including implementation science that you highlight towards the end. Conclusion in this section I am please to see you address the implications of your study as being linked to future study design and implementation. However, you touch on hydro-dilatation which is not mentioned in the discussion (although I would agree it does come up and is relevant in the results). I would suggest if it is an important finding and subsequent recommendation to evaluate it should be raised in the discussion.
--	--

VERSION 1 – AUTHOR RESPONSE

Reviewer: 1

Reviewer Name: Giovanni Galeoto

Institution and Country: Sapienza University of Rome

Please state any competing interests or state 'None declared': None declared

Comments to the Author

The article is well constructed and consistent with international guidelines.

- 1) I ask the authors to cite a work that could be useful for the introduction and discussion of the manuscript:
Cavaliere, E., Servadio, A., Berardi, A., Tofani, M., & Galeoto, G. (2020). The Effectiveness of Physiotherapy in Idiopathic or Primary Frozen Shoulder: a Systematic Review and Meta-Analysis. *Muscles, Ligaments & Tendons Journal (MLTJ)*, 10 (1).

Author: We thank you for your feedback. The above citation is included as an additional reference (Reference 5) in the introduction of the main text.

Reviewer: 2

Reviewer Name: Glen Rae

Institution and Country: Sunderland and South Tyneside Foundation Trust, UK

Please state any competing interests or state 'None declared': Nil

Comments to the Author
None

Reviewer: 3

Reviewer Name: Dr Claire Diver

Institution and Country: School of Health Sciences

University of Nottingham, England, UK

Please state any competing interests or state 'None declared': None declared

Comments to the Author

Whilst I have some points to raise that I think need addressing I found this study interesting, relevant and well written. I would support a re-submission.

Abstract

- 1) The methods in the abstract do not detail the method of data analysis.

Author: Thank you. Please see details added under 'Design' section as below:

'We used constant comparison method to develop our themes'

- 2) The results in the abstract and main body of text report on participant perceptions; if perceptions are a phenomena to be explored then this should be reflected in the title and study aims as perceptions are a different phenomena to experiences.

Author: Thank you. We agree to indicate both experiences and perceptions and report them consistently in our study title, abstract and full-text.

- 3) The conclusion in the abstract does not state what 'important insights' have been gained and as it stands lacks specificity to this study.

Author: Thank you. We have now stated the important messages from our interview findings as below:

'In general, our findings indicated that trial participants, and surgeons and physiotherapists who delivered the treatments had positive experiences and perceptions in the UK FROST. Early qualitative investigations to explore the feasibility of delivering treatments in real-world settings are suggested in future trials in frozen shoulder.'

Introduction

- 1) Does not adequately provide the gap in the literature or rationale for this study of FROST trial experience.

Author: Thank you. We have addressed this in Introduction section as below:

'The qualitative literature on frozen shoulder is limited [11, 12]. There are no trials in frozen shoulder with an embedded qualitative component that explored the perspectives of trial participants (patients with frozen shoulder) and healthcare professionals who delivered the trial treatments.'

- 2) p6 line 19 states little is known about the patient experience of frozen shoulder; this study however does not fill this gap as it is focussed on trial experience rather than condition experience.

Author: Thank you. The sentence has been removed.

- 3) pg6 line 29 states this was a study to look at experience and ACCEPTABILITY. The authors would be encouraged to be consistent throughout including the title, abstract and full text as to the aim of this study.

Author: Thank you. We have now consistently used 'perceptions and experiences' in our study title, abstract and full-text.

Methods

- 1) pg6 line 40. Patients with a diagnosis of frozen shoulder were included in the study; can you state what the diagnostic criteria was for frozen shoulder as this is not given

Author: Thank you. We have included the clinical diagnostic criteria for frozen shoulder as below:

'Participants who took part in the main trial had primary unilateral frozen shoulder, identified through a restriction of passive external rotation on clinical examination.'

- 2) pg 4 line 42. It is not clear why participants were recruited at the end of the trial rather than during it; can you justify.

Author: Thank you. We have justified the time point for conducting the interviews as below:

'A purposive sample of trial participants was recruited from those who had agreed to be contacted by the trial team for interviews at approximately 12 months post-randomisation, which coincided with the primary endpoint of the trial and completion of UK FROST treatments [10].'

- 3) pg6 line 46. There is no justification for the sample size e.g. pragmatic, data saturation or information power. Can this be added. nor the method of sampling used e.g. purposive, convenience etc

Author: Thank you. We have added more details in Methods as below:

'A purposive sample of trial participants was recruited from those who had agreed to be contacted by the trial team for interviews at approximately 12 months post-randomisation, which coincided with the primary endpoint of the trial and completion of UK FROST treatments [10]. We proposed to recruit up to 45 trial participants. We also planned to recruit a purposive sample of up to 15 surgeons and physiotherapists who delivered the treatments in the trial and agreed to take part in the interviews.'

- 4) pg 6 line 50; the professional identity of the researcher conducting the interviews is not stated; this could affect the data collected and its analysis

Author: Thank you. The professional identity of researcher who analysed the interviews has been added as below:

'The primary author (CS) is a physiotherapy researcher trained in qualitative methods and not involved in the delivery of UK FROST treatments and outcome measurement. CS conducted all the interviews and led the analysis. FT is an experienced qualitative researcher, anthropologist and physiotherapist, also not involved in delivery of care or outcome measurement. FT played a collaborative role in analysis.'

- 5) pg6 line 52 there is no justification for the use of a qualitative methodological approach or the use of semi-structured interviews. Can this be added.

Author: Thank you. The selection of qualitative approach has been justified as below:

'A qualitative and constructivist approach [13] was adopted to understand the experiences and perceptions of trial participants and healthcare professionals in UK FROST.'

- 6) it is not clear whether there was a separate interview guide for patient participants and the health care professionals. This needs stating.

Author: Thank you. We used separate interview guides and addressed in Methods as below:

'Separate semi-structured interview guides with open-ended questions were used for trial participants and health professionals.'

- 7) p6 line 54 It is not stated who transcribed the interviews; for transparency can this be stated.

Author: Thank you. We have addressed the above comment as below:

'Interviews were audio-recorded using a digital recorder, anonymised and transcribed verbatim by a professional transcription agency.'

- 8) There is no mention of an information sheet being provided to participants or how the approach to health care professionals was made.

Author: Thank you. We have added more details as below:

'The study information sheet and consent form were posted to potential trial participants and emailed to surgeons and physiotherapists. Two postal or email reminders were sent to those who didn't respond within four weeks of invitation. Once signed consent was received via post or email from volunteer participants, surgeons and physiotherapists, CS contacted potential participants via telephone to coordinate an interview appointment with them.'

Results

- 1) Throughout the results the description of the theme is limited. I would expect to see this expanded and more discrete use of quotes to demonstrate the validity of the description. The multiple quotes for one sub-theme often add little more of value to their presence in the data. I would suggest a re-write of the results with this in mind. The theme 'participation in UK FROST has the level of description I would like to see for each theme; this is in contrast to theme 2 Acceptability of treatment which is dominated by quotes rather than description of the theme.

Author: Thank you. We have added more description as below:

Acceptability - Treatment satisfaction and improved outcomes

All UK FROST treatments were found to be acceptable, satisfactory and beneficial. Except for two participants in the ESP arm, all other participants were satisfied with the UK FROST treatments they received. Of these two exceptions, one did not improve after physiotherapy in the trial and the other was not satisfied when the exercise sessions were supervised by an unfamiliar physiotherapist.

"I'm absolutely delighted with the treatment that I was given. I feel as though it did everything that I wanted it to do and expected it to do". (ESP)

"Very satisfied. I have no complaints at all" (ACR).

Participants considered pain relief and return of shoulder movements and function as important treatment outcomes. Participants in all arms experienced improvements in pain, movements and function. In spite of achieving pain relief and improved function, some participants said they experienced mild and occasional pain and restrictions during certain end-range activities. However, this did not impact their daily functioning.

Pain:

Trial participants experienced a substantial reduction in pain in all treatment arms. The ESP arm participants said that the steroid injections reduced pain and allowed them to start physiotherapy.

“When I went to the surgeon I was injected into my shoulder and the pain down my arm that more or less went straightaway.” (ESP)

“So at the beginning I said the pain was ten and now after all my physios, I’d say it was, I’d say it was about two now.” (MUA)

“I mean the pain in the beginning was just horrendous, it was really, really sore, really painful but after I’d had the physiotherapy, it was... I’ve got no pain at all now.” (ACR)

Movements:

Trial participants described how the physiotherapy sessions (ESP and post-procedural) had helped to improve their shoulder movements.

“I could tell initially straightaway that my movement was starting to come, within a few days I could tell a difference of doing the exercises and as the weeks went on, it was just got better and better and by the time the twelve weeks was up, I virtually had full movements with no pain or anything, it was brilliant!” (ESP)

“After a few days I was doing my exercises and I was quite surprised already how much movement I had back and then it was regular physio appointments up at the hospital just to keep moving things around and that went really well...the physiotherapy was actually really, really beneficial.” (MUA)

In the ACR arm, improvement in movements was thought to be quicker than participants had expected. Some experienced improvements as early as one to two weeks of physiotherapy after surgery.

“I felt that the physiotherapy I received was marvellous and improved the range of movements or showed me how to keep that range of movements much quicker than they did on the right-hand side, so I felt that everything went along fine, and I’ve got no complaints at all, none.” (ACR)

“It is almost like you have had a quick fix to fix your shoulder then you move on and I think personally for me because the surgery went very well and almost after a couple of weeks I was back to normal.” (ACR)

Function:

Trial participants in all treatment arms described how their ability to do routine activities improved.

“I can lift my arm above my head now, you know? I can carry stuff, and I can lift it above my waist, and I can actually go swimming, you know? I can swim now.” (ESP)

"My little everyday things have come back; I have come back, yes". (MUA)

"I can do everything – there's nothing that I can't do; I can wash my back, I can put my bra on, fasten it at the back, I can fasten my skirt at the side and the back now, there's nothing I can't do before I had the frozen shoulder everything I could do then I can now do again".
(ACR)

- 2) p8 Theme 1 Living with frozen shoulder. The aim of the study is to explore the experiences of trial participation. This theme does not do this, but touches on reasons for trial participation. I would suggest this theme is therefore removed as you describe trial participation as a separate theme.

Author: Thank you. We appreciate your feedback on this topic. We have now removed the whole theme on living experiences and the supplementary file on participant narratives.

- 3) p11 line 46 'a stage based approach to routine clinical practice'. As with theme 1 for patient participants this theme does not describe the experiences of delivering treatment in the context of the FROST trial. I would suggest that this is therefore removed and the emphasis is on the delivery of treatment in the context of how it was delivered in FROST, assuming you have the data. This would have a slightly emphasis than theme 2 for this group ie theme 1 is about delivery of treatment in FROST and theme 2 about their expectations and preferences.

Author: We agree with your comments and removed the theme on routine practice. We have added a theme and described in detail as below:

Experiences and perceptions of delivering UK FROST treatments

In general, surgeons and physiotherapists had positive experiences in delivering the trial treatments. Surgeons described how the surgical treatments were no different from their usual practice. They described how trial participants who received MUA or ACR were usually seen by specialised physiotherapists for advice and PPP at the follow-up stage. Some said there were no reports of complications or side-effects due to surgery, whilst some said that a few patients did not respond well to ESP.

Surgeon 01: *"Technically there was no difference, I wouldn't do anything different".*

Surgeon 06: *"In the physiotherapy group, there have been a couple of patients who have not responded very well to surgery, very well to the treatment, to the physiotherapy treatment and we have just carried on and they've taken a bit longer."*

Physiotherapists felt that the UK FROST physiotherapy programmes and the exercise booklet gave more flexibility in choosing the exercises than they would usually have in their routine practice. The only difference was the increased number of sessions offered in the trial. Surgical soreness was a commonly seen post-operative issue in ACR and MUA participants. However, it was quick to resolve with PPP compared to those who received ESP. There were a few suggestions from physiotherapists to improve the UK FROST exercise intervention. For example, one physiotherapist felt the exercises could have been standardised better. Two others suggested either spreading the 12 weekly sessions over 6 months or doing group sessions.

Physiotherapist 1: *"I think the interventions that were on the booklet were what I would use generally. There was always an option there for me to tick off what I would do so I was in agreement with the options that were there and in agreement with the options that they actually, didn't want you to use".*

Physiotherapist 5: *"I think group sessions would be really useful because patients get a lot from each other, and having experience group sessions with other clients with different pathologies, you know, they find that really reassuring..."*

Physiotherapists also commented on the feasibility of delivering the UK FROST physiotherapy programmes within the NHS. There was a sense that it would be difficult to deliver the number of UK FROST physiotherapy sessions in routine practice.

Physiotherapist 4: *"... they (Trial participants) were seen with the 24 hours' post-surgery and they had twelve sessions which is a luxury because in our Trust, that is never, not going to happen and that never used to happen"*.

Discussion

- 1) As with the results there is a discussion of living with frozen shoulder that is outside the aim of the study which is to understand the experiences of participation in the trial itself. I would therefore suggest this is re-written and there is less emphasis on living with frozen shoulder and treatment in routine practice. Some conclusions are drawn that I do not see embedded in the results e.g. steroids help i can only seen mentioned once in patients theme 2 where receiving an injection is mentioned, and is in health care professionals approach in routine clinical practice rather than in the context of this trial

Author: We have ensured that descriptions that are out of the context of UK FROST (Living experiences and Routine practice patterns) are removed throughout the document.

In Results section, we would retain our description about steroid injections as it came out prominently from our interviews with trial participants. To avoid confusion of reporting findings that are out of context (routine practice), we removed the paragraph on steroids in Discussion from our previous version.

Strengths and Limitations

- 1) You acknowledge that findings cannot be extrapolated outside of the context of the FROST trial and yet some of your findings and areas for discussion are indicative of phenomena that occur outside of the trial eg routine practice, and living with frozen shoulder. If this study is about the FROST trial I would suggest that the emphasis remains there. This also links with: implications for practice. I feel that with the aim of the study to look at experiences of participating in the FROST trial the implications should be less about clinical practice and more about future research design including implementation science that you highlight towards the end.

Author: We have ensured that the study findings are relevant within the context of UK FROST throughout the document.

Implications for practice and research: We highlighted only the key findings regarding UK FROST treatments, and focused more on Implications of research as below:

'Implications for clinical practice

1. All UK FROST treatments were perceived as acceptable, beneficial, and satisfactory. The benefits and anticipated risks of these treatments must be clearly communicated to patients during shared treatment decision-making.

Implications for future research

1. Trial participants and healthcare professionals have preferences for treatments. Future studies should aim to understand how these preferences may influence trial results.
2. Implementation of UK FROST physiotherapy programmes within different NHS settings may be challenging. Pilot implementation studies to identify scaling-up strategies would be useful.
3. More large-scale and high-quality randomised controlled trials to ensure clinical effectiveness and safety of hydrodilatation are required to guide evidence-based practice.'

Conclusion

- 1) In this section I am pleased to see you address the implications of your study as being linked to future study design and implementation. However, you touch on hydro-dilatation which is not mentioned in the discussion (although I would agree it does come up and is relevant in the results). I would suggest if it is an important finding and subsequent recommendation to evaluate it should be raised in the discussion.

Author: Hydro-dilatation came up as an important finding from our interviews with UK FROST surgeons and physiotherapists. We have now highlighted in both Discussion and in Implications for future research sections as below:

'Discussion: Surgeons and physiotherapists also perceived hydrodilatation as easy to administer, less invasive and a cost-effective treatment for frozen shoulder instead of surgery. These findings resonate with its growing popularity and usage in clinical practice [31], in spite of a lack of sufficient evidence on efficacy and safety [32, 33].

Implications for research: More large-scale and high-quality randomised controlled trials to ensure clinical effectiveness and safety of hydrodilatation are required to guide evidence-based practice.'

VERSION 2 – REVIEW

REVIEWER	Dr Claire Diver School of Health Sciences University of Nottingham England UK
REVIEW RETURNED	18-Mar-2021
GENERAL COMMENTS	The authors should be commended on addressing the recommendations from the previous submission. I have a few minor comments: Abstract. The setting is described as interviews. This is the method of data collection. I would suggest changing this to where the STUDY was set. There is no mention of reflexivity to enhance credibility of the study: I would suggest it is added if it was used. p6/19 line 2: you state a formal theory was not developed. Can you explain why?

VERSION 2 – AUTHOR RESPONSE

Reviewer: 3: Dr. Claire Diver, University of Nottingham

Comments to the Author: The authors should be commended on addressing the recommendations from the previous submission. I have a few minor comments:

1) Reviewer: Abstract. The setting is described as interviews. This is the method of data collection. I would suggest changing this to where the STUDY was set.

Author's response: Thank you for your suggestion. We have changed the study setting as below.

Setting: ~~Face-to-face or telephone interviews.~~ This qualitative study was nested within the UK FROST. Please see Abstract, Page 2, and Line 16.

2) Reviewer: There is no mention of reflexivity to enhance credibility of the study: I would suggest it is added if it was used.

Author's response: We have added another reflexive statement as we agree that this is an important aspect of credibility. The following sentence is now added to the Discussion section, see Page 14 under 'Strengths and Limitations' lines 39-41: 'We also discussed our preliminary findings with a team of physiotherapists who treat people with frozen shoulder and the UK FROST team to produce our final themes.'

We would like to signpost on the reflexive statements that have been already addressed in the manuscript as below:

Please see Methods section, Page 4, lines 38-42: 'The primary author (CS) is a physiotherapy researcher trained in qualitative methods and not involved in the delivery of UK FROST treatments and outcome measurement. CS conducted all the interviews and led the analysis. FT is an experienced qualitative researcher, anthropologist and physiotherapist, also not involved in delivery of care or outcome measurement. FT played a collaborative role in analysis.'

The credibility of our study findings was enhanced by clarifying the codes and categories and debriefing on the interview dynamics with an experienced qualitative researcher through regular meetings during all stages of analysis. This can be seen in the Discussion section, Page 14 under 'Strengths and Limitations' lines 36-39: 'Interviews were conducted by a researcher not involved in UK FROST and by using open-ended questions that allowed interviewees to express their opinions freely. The interview codes and themes were reviewed by an experienced qualitative researcher to ensure rigour of analysis and interpretation of data.'

3) Reviewer: P6/19 line 2: you state a formal theory was not developed. Can you explain why?

Author's response: Our aim was to explore participants' experience of taking part in the trial and to develop themes that help us to understand and articulate this experience, rather than to develop social theory. We have therefore removed the sentence that refers to formal theory.